# LARGE SPARSE KERNELS FOR FEDERATED LEARNING

**Feilong Zhang [1] \*, Yinchuan Li [2] \*, Shiyi Lin [1], Yunfeng Shao [2], Junjun Jiang [1], Xianming Liu [1] ✉**

[1] Harbin Institute of Technology, Harbin
[2] Huawei Noah's Ark Lab, Beijing, China
Corresponding authors: `csxm@hit.edu.cn`

## ABSTRACT

Existing approaches to address non-iid data in federated learning are often tailored to specific types of heterogeneity and may lack generalizability to all scenarios. In this paper, we present empirical evidence supporting the claim that employing large sparse convolution kernels can lead to enhanced robustness against distribution shifts in the context of federated learning for various non-iid problems, including imbalanced data volumes, different feature spaces, and label distributions. Our experimental results demonstrate that the substitution of convolutional kernels with large sparse kernels can yield substantial improvements in the ability to resist non-iid problems across multiple methods.

## 1 INTRODUCTION

Federated learning (FL) (McMahan et al., 2017; Zhang et al., 2023) has gained popularity as a collaborative machine learning paradigm on decentralized data, enabling privacy-preserving collaboration among multiple parties. However, due to factors such as different label distribution (Gao et al., 2022b; Lin et al., 2023), unbalanced data volume (Li et al., 2020a), and different feature spaces (Jiang et al., 2022), the non-iid nature of the data (Li et al., 2020b) brings challenges to FL (Gao et al., 2022a). In addition to using personalized algorithms (Zhang et al., 2022) to solve this problem, tailoring the structure of the network for a specific type of heterogeneity is also a common approach, but these architectural approaches usually lacks generality (Li et al., 2022) for all scenarios. Therefore, a general network architecture scheme that can be used for all forms of non-iid data is needed, since various forms of heterogeneity can coexist in practice.

In this paper, we propose a novel approach to address non-iid data in FL by leveraging large sparse convolutional kernels, named FedKernel. By increasing the receptive range of the model and providing a focusing effect similar to that of the human eye, our method enhances the transferability of the network and enables it to handle all forms of non-iid data. **Main contributions:** 1) To the best of our knowledge, we are the first to discover that the non-iid problem in FL can be solved by heterogeneous convolution kernel structures; 2) The proposed method is an FL architecture design concept, which can be combined with various non-iid FL algorithms; 3) Experimental results show that the proposed method outperforms existing methods in various non-iid types.

## 2 METHOD

To enhance the accuracy of a model, we formulate the problem as follows:

$$\text{accuracy}(\text{ model }) = f(\mathcal{A}, \mathcal{T}, \mathcal{N}), \tag{1}$$

where $\mathcal{A}$ denotes the architecture design, $\mathcal{T}$ represents the training strategy, and $\mathcal{N}$ indicates the measurement noise that may arise from data heterogeneity among nodes in federated learning. As reducing $\mathcal{N}$ directly is not feasible due to privacy constraints, most existing methods modify the training strategy $\mathcal{T}$ while neglecting the model architecture $\mathcal{A}$. However, training strategies usually only address specific instances of data heterogeneity and may not be effective for all instances of heterogeneity. Therefore, we focus on the model design, starting from the fundamental concept of receptive fields. Intuitively, larger receptive fields can provide better transferability to the model.

---

*Equal Contribution

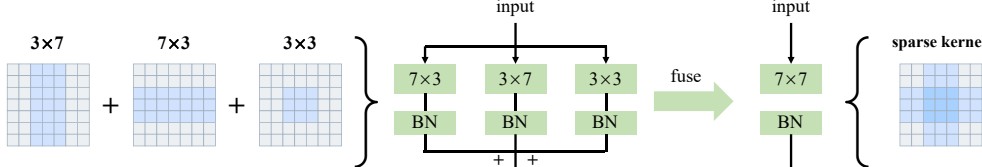

Figure 1: The overview of sparse focus kernel.

To achieve this goal, one approach is to increase the size of the convolutional kernel. However, directly increasing the kernel size would result in a quadratic increase in the number of parameters. Therefore, a sparse reparameterization method is proposed, involving the use of three parallel convolutional kernels $\boldsymbol{F}^{(1)}, \boldsymbol{F}^{(2)}, \boldsymbol{F}^{(3)}$ with sizes $M \times N$, $N \times M$, and $N \times N$, where $M > N$. As shown in Fig. 1, during the training phase, the weights of these kernels are learned. During the testing phase, the three kernels are combined into a single $M \times M$ convolutional kernel using the following equation:

$$\boldsymbol{F}' \leftarrow \boldsymbol{F}^{(1)} + \boldsymbol{F}^{(2)} + \boldsymbol{F}^{(3)}, \quad \boldsymbol{b}' \leftarrow \boldsymbol{b}^{(1)} + \boldsymbol{b}^{(2)} + \boldsymbol{b}^{(3)}, \tag{2}$$

where $\boldsymbol{b}$ represents the bias, and the weights of the central region of the merged kernel are naturally larger than those of the other regions due to the parallel structure. This mimics the focusing effect of the human eye and enhances the CNN's ability to capture important features in the input image. By employing the sparse convolution of large kernels, the proposed method effectively increases the receptive field to improve the network's transferability, enabling it to perform well in the face of various data heterogeneity problems.

## 3 EXPERIMENTS

We evaluate the proposed FedKernel on three different non-iid settings, including label non-iid, feature non-iid, and quantity non-iid. The experiments are conducted on the mini-ImageNet dataset, using the ResNet34 model as the base architecture. Tab. 1 summarizes the test accuracy of FedKernel and four baseline methods: FedAvg, FedProx (Li et al., 2020b), Moon (Li et al., 2021) and HarmoFL (Jiang et al., 2022), on each non-iid setting.

| Method | Category | | |
|---|---|---|---|
| | Label non-iid $p_k \sim \text{Dir}(0.5)$ | Feature non-iid $\hat{\mathbf{x}} \sim \text{Gau}(0.1)$ | Quantity non-iid $q \sim \text{Dir}(0.5)$ |
| FedAvg | 64.37 | 65.17 | 69.21 |
| + FedKernel | 65.36 (+0.99) | 67.13 (+1.96) | 69.93 (+0.72) |
| FedProx | 65.13 | 64.83 | 69.33 |
| + FedKernel | 66.37 (+1.24) | 66.39 (+1.56) | 69.69 (+0.36) |
| Moon | 66.91 | 65.16 | 64.19 |
| + FedKernel | 67.68 (+0.77) | 67.63 (+2.47) | 65.57 (+1.38) |
| HarmoFL | 64.64 | 67.31 | 70.37 |
| + FedKernel | 65.89 (+1.25) | 69.17 (+1.86) | 70.91 (+0.54) |

Table 1: Comparison Results.

The results show that FedKernel consistently outperforms the baselines on all non-iid settings. Specifically, on label non-iid, FedKernel achieves an average improvement of 1.11%, 2.69%, and 1.77% compared to FedAvg, FedProx, and Moon, respectively. On feature non-iid, FedKernel achieves an average improvement of 1.96%, 1.26%, and 2.47%, respectively. On quantity non-iid, FedKernel achieves an average improvement of 0.72%, 0.36%, and 1.38%, respectively. This demonstrates the generality of the proposed FedKernel approach.

## 4 CONCLUSION & FUTURE WORK

Our paper is to explore the potential of large sparse convolution kernels to enhance the robustness of federated learning models against distribution shifts caused by various non-iid data heterogeneities, such as imbalanced data volumes, differing feature spaces, and abnormal distributions. However, the current experiments only report the performance of 7x7 sparse convolution kernels. Future work is needed to explore the impact of larger kernel sizes on the receptive field and the ability to resist non-iid problems in federated learning. Future research can also explore the scalability and applicability of our proposed method to more complex and diverse non-iid scenarios. The proposed approach has the potential to improve the performance of federated learning models in real-world applications where non-iid data heterogeneities are common.

## ACKNOWLEDGEMENTS

This work was supported by National Natural Science Foundation of China under Grants 92270116 and 62071155.

## URM STATEMENT

The authors acknowledge that the first author of this work meet the URM criteria of the ICLR 2023 Tiny Papers Track. The first author is 27 years old and non-white student.

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

## A APPENDIX

In the appendix, we provide a detailed description of our experimental settings.

**Non-IID setting**: We conduct experiments under three different non-iid settings: label non-iid, feature non-iid, and quantity non-iid. To simulate label non-iid, we use the Dirichlet distribution. Specifically, we sample $p_k \sim \text{Dir}_N(\beta)$ and allocate a proportion, $p_{k,j}$, of the instances of class $k$ to party $j$. Here, $\text{Dir}(\cdot)$ denotes the Dirichlet distribution and $\beta$ is a concentration parameter ($\beta > 0$).

A major advantage of this approach is its flexibility in adjusting the imbalance level by varying the concentration parameter $\beta$, which we set to 0.5. Similarly, for distribution-based label imbalance setting, we use the Dirichlet distribution to allocate different amounts of data samples into each party. We sample $q \sim \text{Dir}_N(\beta)$ and allocate a $q_j$ proportion of the total data samples to party $P_j$. For ease of presentation, we use $q \sim \text{Dir}(\beta)$ to denote this partitioning strategy. To simulate feature non-iid, we add different amounts of Gaussian noise to the datasets of different parties. Specifically, we add noise $x \sim \text{Gau}(\sigma \cdot i/N)$ for party $P_i$, where $\text{Gau}(\sigma \cdot i/N)$ is a Gaussian distribution with mean 0 and variance $\sigma \cdot i/N$. we can adjust $\sigma$ to increase the feature dissimilarity among the parties, here, we set $\sigma$ to 0.5.

