# OpenReview forum: "Large Sparse Kernels for Federated Learning"
_ICLR.cc/2023/TinyPapers — Submitted to Tiny Papers @ ICLR 2023_

### Official Review · Reviewer_xvSf · 2023-03-28

**Confidence:** 3

**Summary Of Contributions:**

This work introduces FedKernel, a sparse convolutional kernel approach for models in the Federated Learning setting, which seems to improve generalisation to distribution shifts and transferability. The work outperforms existing baselines that do not use FedKernel on mini-ImageNet with a ResNet-34

**Rating:**

High Potential (HP): a submission which meets the reviewing criteria and has potential to make an impact on the field

**Strengths And Weaknesses:**

Hello authors, thank you for choosing to participate in Tiny Papers this year. Here are some comments:

**STRENGTHS**
- Clear description of the method and what it brings to the table, especially in the FL vision space
- Evident from the results the impact of FedKernel on generalising across different non-iid settings
- Novel idea that has the potential to become a full paper if more experiments and ablations are included

**Suggested Changes:**

No breaking changes, meets the standard for Tiny Papers.

> **URGENT:** If you haven't already, please include the URM statement in the camera-ready version.

---

### Official Review · Reviewer_B9kR · 2023-04-02

**Confidence:** 4

**Summary Of Contributions:**

This paper explores a new approach called FedKernel to address different heterogeneity in FL. This approach tries to reparameterize the convolution kernel into multiple sparse kernels. This can further enhance the robustness of federated learning.

**Rating:**

High Potential (HP): a submission which meets the reviewing criteria and has potential to make an impact on the field

**Strengths And Weaknesses:**

**Strengths:**

S1: The approach of using sparse kernels for federated learning is novel. And this approach is general enough to be easily adapted into different SOTA FL algorithms (FedAvg, FedProx and Moon etc).

S2. This paper is well written. The logic is easy to follow. All the findings in the paper is expressed clearly and effectively.

S3. Experiment setup (dataset, Non-IID partition, baseline algorithms) is clear and easy to be reproduced. Experimental results are strong enough to cover the statements in the paper.

S4: The claims and conclusions are justified by the findings. Authors follow the formatting requirements.

**Weakness:**

W1: Authors can explore different kernel size and different datasets to make the experiment stronger.  Also, authors should state the number of devices in their experiment setup.

W2: Compare to the normal 7x7 kernel,  why this kind of kernel with combined sparsity (7x3, 3x7, 3x3) can result in better accuracy? Authors could give more explanation about this.

W3:  Authors should release their source code if possible.



**Suggested Changes:**

As stated in weakness, some improvements could be:

C1: as W1, add more details about the experiments.

C2: Please consider to cite the following related work in introduction.

[1] FedDCT: Federated Learning of Large Convolutional Neural Networks on Resource Constrained Devices using Divide and Co-Training

[2] Federated Multiple Label Hashing (FedMLH): Communication Efficient Federated Learning on Extreme Classification Tasks

[3] Federated learning on non-IID data: A survey

[4] Real-world image datasets for federated learning

---

### Author Response · Authors · 2023-05-31
**New Paper Revision**

Dear ICLR Program Chairs and Reviewers:

We have uploaded a revised version of our paper.

We would like to opt-in for archival publication.

Thank you.

Best regards.

---

### Comment · Area_Chair_KgFg · 2023-06-02
**Archival**

This work meets the threshold for archival, contents the URM statement and is deanonymized

---

### Meta-Review · Area_Chair_KgFg · 2023-04-04

**Recommendation:** Invite to present (notable)
**Confidence:** 4

**Metareview:**

This paper introduces FedKernel which is a large sparse convolution kernel approach that can lead to enhanced robustness against distribution shifts in federated learning settings.

**Summary:**

The paper has raised some great points for further research on top of the already presented results. All the reviewers share the excitement.

**Reason For Not Giving A Higher Recommendation:**

n/a

**Reason For Not Giving A Lower Recommendation:**

The presentation quality and the excitement expressed by the reviewers make this a great submission.

---

### Decision · Program_Chairs · 2023-04-10

Invite to present (notable)